# CoDet: Co-Occurrence Guided Region-Word Alignment for Open-Vocabulary Object Detection

**Chuofan Ma**[1][†]   **Yi Jiang**[2][*]   **Xin Wen**[1][*]   **Zehuan Yuan**[2]   **Xiaojuan Qi**[1]

[1]The University of Hong Kong    [2]ByteDance Inc.

## Abstract

Deriving reliable region-word alignment from image-text pairs is critical to learn object-level vision-language representations for open-vocabulary object detection. Existing methods typically rely on pre-trained or self-trained vision-language models for alignment, which are prone to limitations in localization accuracy or generalization capabilities. In this paper, we propose CoDet, a novel approach that overcomes the reliance on pre-aligned vision-language space by reformulating region-word alignment as a co-occurring object discovery problem. Intuitively, by grouping images that mention a shared concept in their captions, objects corresponding to the shared concept shall exhibit high co-occurrence among the group. CoDet then leverages visual similarities to discover the co-occurring objects and align them with the shared concept. Extensive experiments demonstrate that CoDet has superior performances and compelling scalability in open-vocabulary detection, *e.g.*, by scaling up the visual backbone, CoDet achieves 37.0 $AP^m_{novel}$ and 44.7 $AP^m_{all}$ on OV-LVIS, surpassing the previous SoTA by 4.2 $AP^m_{novel}$ and 9.8 $AP^m_{all}$. Code is available at https://github.com/CVMI-Lab/CoDet.

## 1   Introduction

Object detection is a fundamental vision task that offers object-centric comprehension of visual scenes for various downstream applications. While remarkable progress has been made in terms of detection accuracy and speed, traditional detectors [39, 37, 21, 5, 43] are mostly constrained to a fixed vocabulary defined by training data, *e.g.*, 80 categories in COCO [29]. This accounts for a major gap compared to human visual intelligence, which can perceive a diverse range of visual concepts in the open world. To address such limitations, this paper focuses on the open-vocabulary setting of object detection [57], where the detector is trained to recognize objects of arbitrary categories.

Recently, vision-language pretraining on web-scale image-text pairs has demonstrated impressive open-vocabulary capability in image classification [35, 25]. It inspires the community to adapt this paradigm to object detection [27, 51], specifically by training an open-vocabulary detector using region-text pairs in detection or grounding annotations [27]. However, unlike free-form image-text pairs, human-annotated region-text pairs are limited and difficult to scale. Consequently, a growing body of research [59, 27, 53, 15, 28] aims to mine additional region-text pairs from image-text pairs, which raises a new question: *how to find the alignments between regions and words?* (Figure 1a)

Recent studies typically rely on vision-language models (VLMs) to determine region-word alignments, for example, by estimating region-word similarity [59, 27, 15, 28]. Despite its simplicity, the quality of generated pseudo region-text pairs is subject to limitations of VLMs. As illustrated in Figure 1b, VLMs pre-trained with image-level supervision, such as CLIP [35], are largely unaware of localization quality of pseudo labels [59]. Although detector-based VLMs [27, 53] mitigate this issue to some extent, they are initially pre-trained with a limited number of detection or grounding concepts,

---

[†]This work was performed when Chuofan Ma worked as an intern at ByteDance.
[*]Equal contribution.

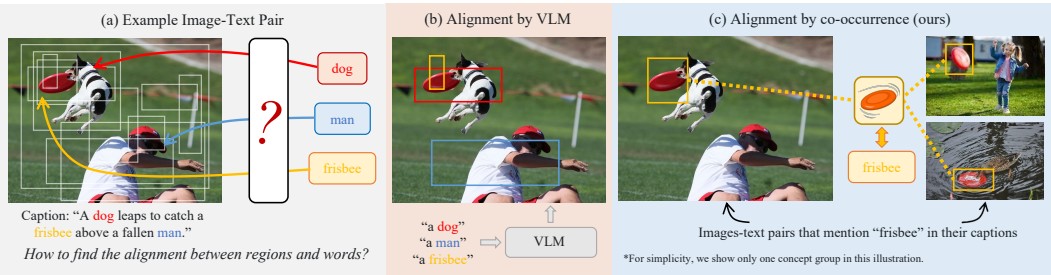

Figure 1: **Illustration of different region-text alignment paradigms.** (a): example image-text pair, and region proposals generated by a pre-trained region proposal network; (b): a pre-trained VLM (*e.g.*, CLIP [35]) is used to retrieve the box with the highest region-word similarity concerning the query text, which yet exhibits poor localization quality; (c) our method overcomes the reliance on VLMs by exploring visual clues, *i.e.*, object co-occurrence, within a group of image-text pairs containing the same concept (*e.g.*, frisbee 🥏). *Best viewed in color.*

resulting in inaccurate alignments for novel concepts [61]. Furthermore, this approach essentially faces a chicken-and-egg problem: obtaining high-quality region-word pairs requires a VLM with object-level vision-language knowledge, yet training such a VLM, in turn, depends on a large number of aligned region-word pairs.

In this work, instead of directly aligning regions and words with VLMs, we propose leveraging region correspondences across images for co-occurring concept discovery and alignment, which we call CoDet. Figure 1c illustrates the idea. Our key motivation is that objects corresponding to the same concept should exhibit consistent visual similarity across images, which provides visual clues to identity region-word correspondences. Based on this intuition, we construct semantic groups by sampling images that mention a shared concept in their captions, from which we can infer that a common object corresponding to the shared concept exists across images. Subsequently, we leverage cross-image region similarity to identify regions potentially containing the common object, and construct a prototype from them. The prototype and the shared concept form a natural region-text pair, which is then adopted to supervise the training of an open-vocabulary object detector.

Unlike previous works, our method avoids the dependence on a pre-aligned vision-language space and *solely relies on the vision space* to discover region-word correspondences. However, there could exist multiple co-occurring concepts in the same group, and even the same concept may still exhibit high intra-category variation in appearance, in which case general visual similarity would fail to distinguish the object of interest. To address this issue, we introduce text guidance into similarity estimation between region proposals, making it concept-aware and more accurately reflecting the closeness of objects concerning the shared semantic concept.

The main contributions of this paper can be summarized as follows:

- We introduce a novel perspective in discovering region-word correspondences from image-text pairs, which bypasses the dependence on a pre-aligned vision-language space by reformulating region-word alignment as a co-occurring object discovery problem.
- Building on this insight, we propose CoDet, an open-vocabulary detection framework that learns object-level vision-language alignment directly from web-crawled image-text pairs.
- CoDet consistently outperforms existing methods on the challenging OV-LVIS benchmark and demonstrates superior performances in cross-dataset detection on COCO and Objects365.
- CoDet exhibits strong scalability with visual backbones - it achieves $23.4/29.4/37.0$ mask $AP_{novel}$ with ResNet50/Swin-B/EVA02-L backbone, outperforming previous SoTA at a comparable model size by $0.8/3.1/4.2$ mask $AP_{novel}$, respectively.

## 2    Related Work

**Zero-shot object detection (ZSD)**    leverages language feature space for generalization to unseen objects. The basic idea is to project region features to the pre-computed text embedding space (*e.g.*, GloVe [34]) and use word embeddings as the classifier weights [2, 10, 36]. This presents ZSD with the flexibility of recognizing unseen objects given its name during inference. Nevertheless,

ZSD settings restrict training samples to come from a limited number of seen classes, which is not sufficient to align the feature space of vision and language. Although some works [63, 42] try to overcome this limitation by hallucinating novel classes using Generative Adversarial Network [17], there is still a large performance gap between ZSD and its supervised counterparts.

**Weakly supervised object detection (WSD)**    exploits data with image-level labels to train an object detector. It typically treats an image as a bag of proposals, and assigns the image label to these proposals through multiple instance learning [4, 9, 3, 46]. By relieving object detection from costly instance-level annotations, WSD is able to scale detection vocabulary with cheaper classification data. For instance, recent work Detic [61] greatly expands the vocabulary of detectors to twenty-thousand classes by leveraging image-level supervision from ImageNet-21K [11]. However, WSD still requires non-trivial annotation efforts and has a closed vocabulary during inference.

**Open-vocabulary object detection (OVD)**    is built upon the framework of ZSD, but relaxes the stringent definition of novel classes from 'not seen' to 'not known in advance', which leads to a more practical setting [57]. Particularly, with recent advancement of vision-language pre-training [35, 25, 55, 54], a widely adopted approach of OVD is to transfer the knowledge of pre-trained vision-language models (VLMs), *e.g.*, CLIP [35], to object detectors through distillation [18, 50] or weight transfer [33, 26]. Despite its utility, performances of these methods are arguably restricted by the teacher VLM, which is shown to be largely unaware of fine-grained region-word alignment [59, 6]. Alternatively, another group of works utilize large-scale image-text pairs to expand detection vocabulary [57, 59, 56, 14, 27, 28, 52], sharing a similar idea as WSD. Due to the absence of regional annotations in image-caption data, these methods typically rely on pre-trained or self-trained VLMs to find region-word correspondences, which are prone to limitations in localization accuracy or generalization capabilities. Our method is orthogonal to all the aforementioned approaches in the sense that it does not explicitly model region-word correspondences, but leverage region correspondences across images to bridge regions and words, which greatly simplifies the task.

**Cross-image Region Correspondence**    is widely explored to discover semantically related regions among a collection of images [20, 41, 45, 44, 30]. Based on the observation that modern visual backbones provide consistent semantic correspondences in the feature space [58, 20, 47, 24, 60, 1], many works take a heuristic approach [41] or use clustering algorithms [20, 8, 23, 49] for common region discovery. Our method takes inspiration from these works to discover co-occurring objects across image-text pairs, with newly proposed text guidance.

## 3    Method

In this section, we present CoDet, an end-to-end framework exploiting image-text pairs for open-vocabulary object detection. Figure 2 gives an overview of CoDet. We first provide a brief introduction to the OVD setup (Sec. 3.1). Then we discuss how to reformulate region-word alignment as a co-occurring object discovery problem (Sec. 3.2), which is subsequently addressed by CoDet (Sec. 3.3). Finally, we summarize the overall training objectives and inference pipelines of CoDet (Sec. 3.4).

### 3.1    Preliminaries

**Task formulation.**    In our study, we adopt the classical OVD problem setup as in OVR-CNN [57]. Specifically, box annotations are only provided for a predetermined set of base categories $\mathcal{C}^{\text{base}}$ during training. While at the test phase, the object detector is required to generalize beyond $\mathcal{C}^{\text{base}}$ to detect objects from novel categories $\mathcal{C}^{\text{novel}}$. Notably, $\mathcal{C}^{\text{novel}}$ is not known in advance to simulate open-world scenarios. This implies the object detector needs to possess the capability to recognize potentially any object based on its name. To achieve this goal, we additionally leverage image-text pairs with an unbounded vocabulary $\mathcal{C}^{\text{open}}$ to extend the lexicon of the object detector.

**OVD framework.**    Our method is built on the two-stage detection framework Mask-RCNN [21]. To adapt it for the open-vocabulary setting, we follow the common practice [18, 61] to decouple localization from classification, by replacing the class-specific localization heads with class-agnostic ones that produce a single box or mask prediction for each region proposal. Besides, to enable

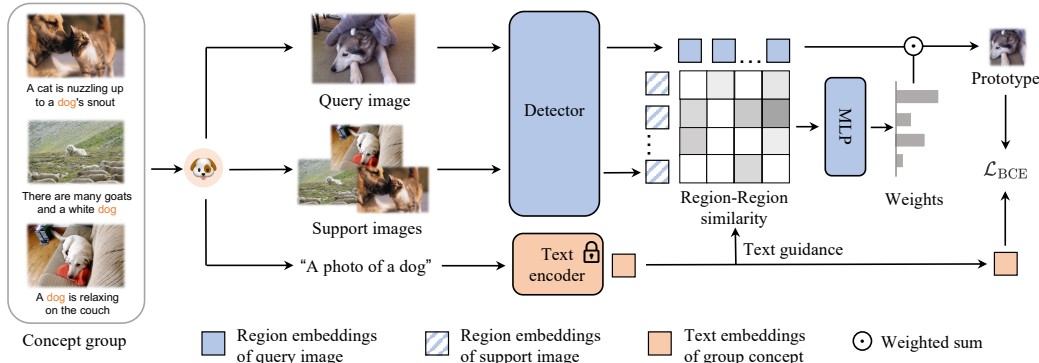

Figure 2: **Overview of CoDet.** Our method learns to jointly discover region-word pairs from a group of image-text pairs that mention a shared concept in their captions (*e.g.*, dog in the figure). We identify the co-occurring objects and the shared concept as natural region-word pairs. Then we leverage inter-image region correspondences, *i.e.*, region-region similarity, with text guidance to locate the co-occurring objects for region-word alignment.

open-vocabulary classification of objects, the fixed classifier weights are substituted with dynamic text embeddings of category names, which are generated by the pre-trained text encoder of CLIP [35].

## 3.2   Aligning Regions and Words by Co-occurrence

Due to the absence of box annotations, a core challenge of utilizing image-text pairs for detection training is to figure out the fine-grained alignments between regions and words. To put it formally, for an image-text pair $\langle I, T \rangle$, $R = \{r_1, r_2, ..., r_{|R|}\}$ denotes the set of regions proposals extracted from image $I$, and $C = \{c_1, c_2, ..., c_{|C|}\}$ denotes the set of concept words extracted from caption $T$. Under weak caption supervision, we can assume the presence of a concept $c$ in $T$ indicates the existence of at least one region $r$ in $I$ containing the corresponding object. However, the concrete correspondence between $c$ and $r$ remains unknown.

To solve the problem, we propose to explore the global context of caption data and align regions and words via co-occurrence. Specifically, for a given concept $c$, we construct a concept group $G$ comprising all image-text pairs that mention $c$ in their captions. This grouping effectively clusters images that contain objects corresponding to concept $c$. Consequently, if $G$ is large enough, we can induce that the objects corresponding to concept $c$ will be the most common objects among images in $G$. This natural correlation automatically aligns the co-occurring objects in $G$ to concept $c$. We thus reduce the problem of modeling cross-modality correspondence (region-word) to in-modality correspondence (region-region), which we address in the next section.

## 3.3   Discovering Co-occurring Objects across Images

Intuitively, candidate region proposals containing the co-occurring object should exhibit consistent and similar visual patterns across the images. We thus take a similarity-driven approach to discover these proposals. As illustrated in Figure 2, during training, we only sample a mini-group of images from the concept group as inputs in consideration of efficiency. In the mini-group, we *iteratively* choose one image as query image, and leave the rest as support images. Note that the regional proposals for each image are cached to avoid re-computation when swapping query and support images. The basic idea is to discover co-occurring objects in the query image from region proposals that have close neighbors across the support images. To fulfill this purpose, we introduce text-guided similarity estimation and similarity-based prototype discovery in the following paragraphs.

**Similarity-based prototype discovery.**    Since modern visual backbones provide consistent feature correspondences for visually similar regions across images [58, 20, 47, 24, 60, 1], a straightforward way to identify co-occurring objects is to measure the cosine similarity between features of region proposals. Concretely, we calculate the pairwise similarity of region proposals between the query image and the support images. This yields a similarity matrix $\mathbf{S} \in \mathbb{R}^{n \times mn}$, where $n$ stands for the number of proposals per image, and $m$ stands for the number of support images. Intuitively,

co-occurring regions should exhibit high responses (similarities) in the last dimension of $\mathbf{S}$. But instead of using hand-crafted rules as in [41], we employ a two-layer MLP, denoted as $\Phi$, to derive co-occurrence from $\mathbf{S}$. $\Phi$ is trained to estimate the probability of each region proposal in the query images as a co-occurring region, solely conditioned on $\mathbf{S}$. Here, we do not explicitly supervise the output probability since there is no ground-truth annotation, but $\Phi$ is encouraged to assign high probabilities for co-occurring regions to minimize the overall region-word alignment loss. Based on the estimated probability vector $\mathbf{p} \in \mathbb{R}^n$, we obtain the prototypical region features for the co-occurring object via simple weighted sum:

$$\mathbf{f}_p = \sum_{i=1}^{N} \mathbf{p}_i \cdot \mathbf{f}_i, \quad \text{where } \mathbf{p} = \operatorname*{softmax}_{N} \left( \Phi \left( \mathbf{S} \right) \right). \tag{1}$$

As the text label for this prototype naturally corresponds to the shared concept $c$ in the mini-group. We can thus learn region-word alignment with a binary cross-entropy (BCE) classification loss:

$$\mathcal{L}_{\text{region-word}} = \mathcal{L}_{\text{BCE}}(\mathbf{W}\mathbf{f}_p, c), \quad \text{where } \mathcal{L}_{\text{BCE}}(\mathbf{s}, c) = -\log \sigma \left( \mathbf{s}_c \right) - \sum_{k \neq c} \log \left( 1 - \sigma \left( \mathbf{s}_k \right) \right), \tag{2}$$

where $\mathbf{W}$ the classifier weight derived from $\mathcal{C}^{\text{open}}$, and $\sigma(\cdot)$ stands for the sigmoid function.

**Text-guided region-region similarity estimation.** However, general cosine similarity may not always truly reflect closeness of objects in the semantic space, as objects of the same category may exhibit significant variance in appearance. Moreover, there could exist multiple co-occurring concepts among the sampled images, which incurs ambiguity in identifying co-occurrence. To address the problems, we introduce text guidance into similarity estimation to make it concept-aware. Concretely, given features of two region proposals $\mathbf{f}_i, \mathbf{f}_j \in \mathbb{R}^d$, where $d$ is the dimension of feature vectors, we additionally introduce $\mathbf{w}_c \in \mathbb{R}^d$, the text embedding of concept $c$ (the shared concept in the mini-group), to re-weight similarity calculation:

$$s_{ij} = \bar{\mathbf{w}}_c^\top \cdot \left( \frac{\mathbf{f}_i}{\|\mathbf{f}_i\|} \circ \frac{\mathbf{f}_j}{\|\mathbf{f}_j\|} \right), \quad \text{where } \bar{\mathbf{w}}_c = \sqrt{d} \frac{|\mathbf{w}_c|}{\|\mathbf{w}_c\|}, \tag{3}$$

where "$\circ$" denotes Hadamard product, "$|\cdot|$" denotes absolute value operation, and "$\|\cdot\|$" denotes $\ell_2$-normalization, respectively. Here, the rationale is that the relative magnitude of text features at different dimensions indicates their relative importance in the classification of concept $c$. By weighting the image feature similarities with the text feature magnitudes, the similarity measurement can put more emphasis on feature dimensions that reflect more of the text features. Therefore, the re-weighted similarity metric provides a more nuanced and tailored measure of the proximity between objects in the context of a particular concept. It is noteworthy that we choose regional features from the output of the penultimate layer of the classification head (the last layer is the text embedding) so that they naturally reside in the shared feature space as text embeddings.

### 3.4 Training and Inference

Following [61, 28], we train the model simultaneously on detection data and image-text pairs to acquire localization capability and knowledge of vision-language alignments. In addition to learning region-level alignments from region-word pairs discovered in Sec. 3.3, we treat image-text pairs as a generalized form of region-word pairs to learn image-level alignments. Particularly, we use a region proposal covering the entire image to extract image features, and encode the entire caption into language embeddings. Similar to CLIP [35], we consider each image and its original caption as a positive pair and other captions in the same batch as negative pairs. We then use a BCE loss similar to Eq. (2) to calculate the image-text matching loss $\mathcal{L}_{\text{image-text}}$. The overall training objective for this framework is:

$$\mathcal{L}(I) = \begin{cases} \mathcal{L}_{\text{rpn}} + \mathcal{L}_{\text{reg}} + \mathcal{L}_{\text{cls}}, & \text{if } I \in \mathcal{D}^{\text{det}} \\ \mathcal{L}_{\text{region-word}} + \mathcal{L}_{\text{image-text}}, & \text{if } I \in \mathcal{D}^{\text{cap}} \end{cases}, \tag{4}$$

where $\mathcal{L}_{\text{rpn}}, \mathcal{L}_{\text{reg}}, \mathcal{L}_{\text{cls}}$ are standard losses in the two-stage detector. For inference, CoDet does not require cross-image correspondence modeling as in training. It behaves like a normal two-stage object detector by forming the classifier with arbitrary language embeddings.

# 4 Experiments

## 4.1 Benchmark Setup

**OV-LVIS** is a general benchmark for open-vocabulary object detection, built upon LVIS [19] dataset which contains a diverse set of 1203 categories of objects with a long-tail distribution. Following standard practice [18, 59], we set the 866 common and frequent categories in LVIS as base categories, and leave the 337 rare categories as novel categories. Besides, we choose CC3M [40] which contains 2.8 million free-from image-text pairs crawled from the web, as the source of image-text pairs. The main evaluation metric on OV-LVIS is the mask AP of novel (rare) categories.

**OV-COCO** is derived from the popular COCO [29] benchmark for evaluation of zero-shot and open-vocabulary object detection methods [2, 57]. It splits the categories of COCO into 48 base categories and 17 novel categories, while removing the 15 categories without a synset in the WordNet [32]. As for image-caption data, following existing works [57, 61, 28], we use COCO Caption [7] training set which provides 5 human-generated captions for each image for experiments on OV-COCO. The main evaluation metric on OV-COCO is the box $AP_{50}$ of novel categories.

## 4.2 Implementation Details

We extract object concepts from the text corpus of COCO Caption/CC3M using an off-the-shelf language parser [32]. Remarkably, we filter out concepts without a synset in WordNet or outside the scope of the 'object' definition (*i.e.*, not under the hierarchy of 'object' synset in WordNet) to clean the extracted concepts. For phrases of more than one word, we simply apply the filtering logic to the last word in the phrase. Subsequently, we remove concepts with a frequency lower than 100/20 in COCO-Caption/CC3M. This leaves 634/4706 concepts for COCO/CC3M.

For experiments on OV-LVIS, unless otherwise specified, we use CenterNet2 [62] with ResNet50 as the backbone, following [61, 28]. For OV-COCO, Faster R-CNN with a ResNet50-C4 [22] backbone is adopted. To achieve faster convergence, we initialize the model with parameters from the base class detection pre-training as in [61, 28]. The batch size on a single GPU is set to 2/8 for COCO/LVIS detection data and 8/32 for COCO Caption/CC3M caption data. The ratio between the detection batch and caption batch is set to 1:1 during co-training. Notably, a caption batch by default contains four mini-groups, where each mini-group is constructed by sampling 2/8 image-text pairs from the same concept group in COCO Caption/CC3M. We train the model for 90k iterations on 8 GPUs.

Table 1: **Comparison with state-of-the-art open-vocabulary object detection methods on OV-LVIS**. Caption supervision means the method learns vision-language alignment from image-text pairs, while CLIP supervision indicates transferring knowledge from pre-trained CLIP. The column 'Strict' indicates whether the method follows a strict open-vocabulary setting.

| Method | Backbone | Supervision | Strict | $AP_{novel}^m$ | $AP_c^m$ | $AP_f^m$ | $AP_{all}^m$ |
|---|---|---|---|---|---|---|---|
| ViLD [18] | RN50-FPN | CLIP | ✓ | 16.6 | 24.6 | 30.3 | 25.5 |
| RegionCLIP [59] | RN50-C4 | Caption | ✓ | 17.1 | 27.4 | 34.0 | 28.2 |
| DetPro [12] | RN50-FPN | CLIP | ✓ | 19.8 | 25.6 | 28.9 | 25.9 |
| OV-DETR [56] | RN50-C4 | Caption | ✗ | 17.4 | 25.0 | 32.5 | 26.6 |
| PromptDet [14] | RN50-FPN | Caption | ✗ | 19.0 | 18.5 | 25.8 | 21.4 |
| Detic [61] | RN50 | Caption | ✗ | 19.5 | - | - | 30.9 |
| F-VLM [26] | RN50-FPN | CLIP | ✓ | 18.6 | - | - | 24.2 |
| VLDet [28] | RN50 | Caption | ✓ | 21.7 | 29.8 | 34.3 | 30.1 |
| BARON [50] | RN50-FPN | CLIP | ✓ | 22.6 | 27.6 | 29.8 | 27.6 |
| CoDet (Ours) | RN50 | Caption | ✓ | **23.4** | 30.0 | 34.6 | 30.7 |
| RegionCLIP [59] | R50x4 (87M) | Caption | ✓ | 22.0 | 32.1 | 36.9 | 32.3 |
| Detic [61] | SwinB (88M) | Caption | ✗ | 23.9 | 40.2 | 42.8 | 38.4 |
| F-VLM [26] | R50x4 (87M) | CLIP | ✓ | 26.3 | - | - | 28.5 |
| VLDet [28] | SwinB (88M) | Caption | ✓ | 26.3 | 39.4 | 41.9 | 38.1 |
| CoDet (Ours) | SwinB (88M) | Caption | ✓ | **29.4** | 39.5 | 43.0 | 39.2 |
| F-VLM [26] | R50x64 (420M) | CLIP | ✓ | 32.8 | - | - | 34.9 |
| CoDet (Ours) | EVA02-L (304M) | Caption | ✓ | **37.0** | 46.3 | 46.3 | 44.7 |

## 4.3 Benchmark Results

Table 1 presents our results on OV-LVIS. We follow a strict open-vocabulary setting where novel categories are kept unknown during training, to ensure we obtain a generic open-vocabulary detector not biased towards specific novel categories. It can be seen that CoDet consistently outperforms SoTA methods in novel object detection. Especially, among the group of methods learning from caption supervision, CoDet surpasses all alternatives which rely on CLIP (RegionCLIP [59], OV-DETR[56]), max-size prior (Detic [61]), or self-trained VLM (PromptDet [14], VLDet [28]) to generate pseudo region-text pairs, demonstrating the superiority of visual guidance in region-text alignment.

In addition, we validate the scalability of CoDet by testing with more powerful visual backbones, *i.e.*, Swin-B [31] and EVA02-L [13]. It turns out that our method scales up surprisingly well with model capacity - it leads to a +6.0/13.6 $AP_{novel}^m$ performance boost by switching from ResNet50 to Swin-B/EVA02-L, and continuously enlarges the performance gains over the second best method with a comparable model size. We believe this is because stronger visual representations provide more consistent semantic correspondences across images, which are critical for discovering co-occurring objects among the concept group.

Table 2 presents our results on OV-COCO, where CoDet achieves the second best performance among existing methods. Compared with the leading method VLDet, the underperformance of CoDet can be mainly attributed to the human-curated bias in COCO Caption data distribution. That is, images in COCO Caption contain at least one of the 80 categories in COCO, which leads to highly concentrated concepts. For instance, roughly 1/2 of the images contain 'people', and 1/10 of the images contain 'car'. This unavoidably incurs many hard negatives for identifying co-occurring objects of interest. Ablation studies on the concept group size of CoDet in Table 5 reveals the same problem. But we believe this would not harm the generality of our method as we show CoDet works well on web-crawled data (*e.g.*, CC3M), which is a more practical setting and can easily be scaled up.

Table 2: **Comparison with state-of-the-art methods on OV-COCO**. [†]: implemented with Deformable DETR [64].

| Method | $AP_{50}^{novel}$ | $AP_{50}^{base}$ | $AP_{50}^{all}$ |
|---|---|---|---|
| OVR-CNN [57] | 22.8 | 46.0 | 39.9 |
| ViLD [18] | 27.6 | 59.5 | 51.3 |
| RegionCLIP [59] | 26.8 | 54.8 | 47.5 |
| Detic [61] | 27.8 | 47.1 | 42.0 |
| OV-DETR [56][†] | 29.4 | 61.0 | 52.7 |
| PB-OVD [15] | 29.1 | 44.4 | 40.4 |
| VLDet | **32.0** | 50.6 | 45.8 |
| CoDet (Ours) | 30.6 | 52.3 | 46.6 |

Table 3: **Cross-datasets transfer detection from OV-LVIS to COCO and Objects365.** [†]: Detection-specialized pre-training with SoCo [48].

| Method | COCO | | | Objects365 | | |
|---|---|---|---|---|---|---|
| | AP | $AP_{50}$ | $AP_{75}$ | AP | $AP_{50}$ | $AP_{75}$ |
| Supervised [18] | 46.5 | 67.6 | 50.9 | 25.6 | 38.6 | 28.0 |
| ViLD [18] | 36.6 | 55.6 | 39.8 | 11.8 | 18.2 | 12.6 |
| DetPro [12][†] | 34.9 | 53.8 | 37.4 | 12.1 | 18.8 | 12.9 |
| F-VLM [26] | 32.5 | 53.1 | 34.6 | 11.9 | 19.2 | 12.6 |
| BARON [50] | 36.2 | 55.7 | 39.1 | 13.6 | **21.0** | 14.5 |
| CoDet (Ours) | **39.1** | **57.0** | **42.3** | **14.2** | 20.5 | **15.3** |

## 4.4 Transfer to Other Datasets

In simulation to detection in the open world, where test data may come from different domains, we conduct cross-dataset transfer detection experiments in Table 3. Specifically, we transfer the open-vocabulary detector trained on OV-LVIS (LVIS base + CC3M) to COCO and Objects365 v1, by plugging in the vocabulary of test datasets without further model fine-tuning. In comparison with existing works, CoDet outperforms the second-best method ViLD [18] which uses a $32\times$ training schedule by 2.0% and 2.1% AP on COCO and Objects365, validating the generalization capability of CoDet across image domains and vocabularies.

## 4.5 Visualization and Analysis

Discovering reliable region-word alignments is critical to learn object-level vision-language representations from image-text pairs. In this section, we investigate different types of alignment strategies that are primarily based on: 1) region-word similarity, *i.e.*, assigning words to regions of the highest similarity [38, 14]; 2) hand-crafted prior, *i.e.*, assigning words to regions of the maximum size [61];

and 3) region-region similarity, *i.e.*, assigning words to regions of the maximum weight, derived by CoDet from region-region similarity matrix (one may refer to Figure 2 for clarity).

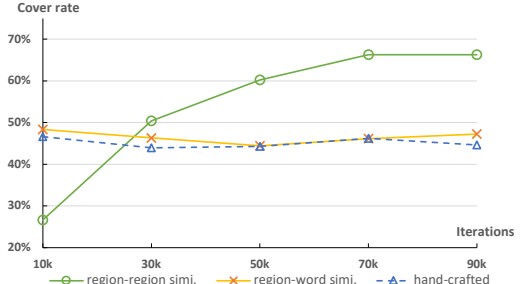

Figure 3: **A comparison of different alignment strategies on OV-COCO.** Cover rate is the ratio of assigned proposal covering the ground-truth box.

Figure 3 shows a comparison of these strategies. Specifically, we employ the model trained with different strategies on OV-COCO benchmark to generate pseudo-labels for novel categories in COCO validation set. Note that the pseudo-labels are generated with the same strategy as in training, not by prediction results. We evaluate the quality of pseudo-labels by cover rate, which is defined as the ratio of pseudo-labels whose assigned region has a mIoU > 0.5 with the closest ground-truth box. At the end of training, we can see that alignments based on region-region similarity produce much more accurate pseudo labels compared with the other two strategies, manifesting the reliability of visual guidance. Another intriguing finding is that, our method can benefit from self-training which leads to steadily increasing pseudo-label quality, while such pattern is not observed in similarly self-trained model relying on region-word similarity for alignment. We conjecture this is because the model gets stuck in the aforementioned chicken-and-egg problem, which is reflected in its unstable training curve (see Appendix B). This also aligns with the finding in [61] [3]. Further qualitative comparisons are presented in Figure 4.

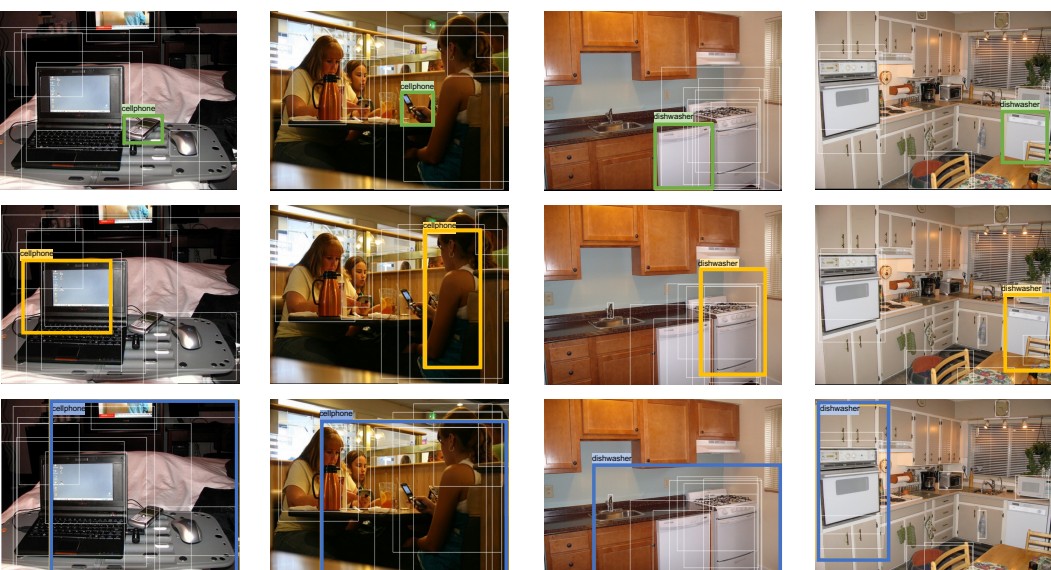

Figure 4: Visualization of pseudo bounding box labels generated by different region-word alignment strategies. From top to bottom, each row shows results of strategies based on region-region similarity, region-word similarity, and hand-crafted prior, respectively. Zoom in for a better view.

## 4.6 Ablation Study

**Text-guidance.** We ablate the impact of text guidance in estimating inter-region similarity on OV-COCO. As shown in Table 4a, introducing text guidance leads to a significant performance boost on novel AP. This finding aligns with our intuition that putting region similarity estimation in a semantic context can provide better measurements of semantic closeness. Figure 5 further demonstrates how text guidance facilitates mitigating interference from irrelevant concepts.

---

[3]Detic shows that matching regions and words based on highest region-word similarity produces highly inconsistent pseudo labels at different training stages.

Table 4: **Ablation study on effective components.** We show that both text guidance and prototype-based strategy substantially facilitate co-occurring object discovery.

(a) **Text guidance**

| Text guide | $AP_{50}^{novel}$ | $AP_{50}^{base}$ | $AP_{50}^{all}$ |
|---|---|---|---|
| ✗ | 26.6 | 52.4 | 45.7 |
| ✓ | 30.6 | 52.3 | 46.6 |

(b) **Strategy for co-occurring object discovery**

| Strategy | $AP_{50}^{novel}$ | $AP_{50}^{base}$ | $AP_{50}^{all}$ |
|---|---|---|---|
| Heuristic [41] | 26.9 | 52.4 | 45.7 |
| Prototype-based | 30.6 | 52.3 | 46.6 |

**Heuristic vs. Prototype-based co-occurring object discovery.** To investigate the effectiveness of prototype-based strategy in co-occurring object discovery, we adopt a heuristic strategy in image co-segmentation [41] for comparison. Results are presented in Table 4b. The heuristic strategy works to select a single region proposal that has close neighbors across support images following hand-crafted rules (please refer to Appendix A for more details). Our prototype-based strategy makes a substantial improvement over this simple baseline by 3.7 AP on novel categories. We speculate the gains mainly come from two aspects: 1) robustness to noisy similarity estimations; We notice that some region proposals in the background may be estimated with high similarity in the early stage, which disturbs the selection by the heuristic strategy. While our prototype-based strategy avoids hard selection by assigning soft weights to each region proposal to construct a prototype, thus is more robust to such noises. 2) the ability to harness multiple instances; considering that there may be multiple instances corresponding to the shared concept in an image, prototype-based strategy can effectively make use of region proposals of different instances to construct a prototype, which we show in Appendix C).

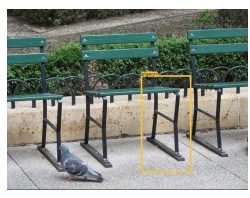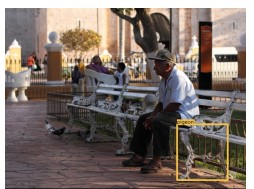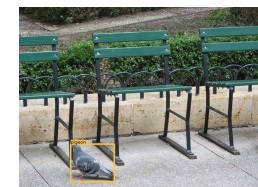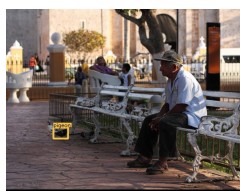

w/o text guidance      w/ text guidance "pigeon"

Figure 5: There can be more than one co-occurring concept among sampled images. Text guidance helps filter out the distracting concept (chair legs) and focus on the concept of interest (pigeons).

**Size of concept group.** In CoDet, co-occurring object discovery is based on a mini-group of images sampled from the same concept group during training. Since the size of the mini-group is generally small, sometimes there could be more than one co-occurring concept in a group, in which case the model will be confused about which concept to discover, as illustrated in Figure 5. Besides introducing text guidance, intuitively, increasing the group size can also effectively reduce this ambiguity, as verified by results on OV-LVIS in Table 5. However, contrary results are observed in experiments on OV-COCO. We speculate these abnormal results are probably caused by the aforementioned human-curated bias in COCO Caption (See Sec. 4.3). Due to the highly concentrated concepts, increasing the group size will undesirably introduce more concurrent concepts that harm the model performances.

Table 5: **Ablation study on concept group size.** CoDet shows different preferences of concept group size on human-curated caption data (OV-COCO) and web-crawled image-text pairs (OV-LVIS).

| Group Size | OV-COCO | | | OV-LVIS | | | |
|---|---|---|---|---|---|---|---|
| | $AP_{50}^{novel}$ | $AP_{50}^{base}$ | $AP_{50}^{all}$ | $AP_{novel}^{m}$ | $AP_{c}^{m}$ | $AP_{f}^{m}$ | $AP_{all}^{m}$ |
| 2 | 30.6 | 52.3 | 46.6 | 21.9 | 30.3 | 35.0 | 30.7 |
| 4 | 29.9 | 51.2 | 45.6 | 21.8 | 30.2 | 34.9 | 30.6 |
| 8 | 29.1 | 50.9 | 45.2 | 22.7 | 30.3 | 34.7 | 30.7 |

# 5    Limitations and Conclusions

In this paper, we make the first attempt to explore visual clues, *i.e.*, object co-occurrence, to discover region-word alignments for open-vocabulary object detection. We present CoDet, which effectively leverages cross-image region correspondences and text guidance to discover co-occurring objects for alignment, achieving state-of-the-art results on various OVD benchmarks. On the other hand, our method is orthogonal to previous efforts in aligning regions and words with VLMs. Combining the advantages of both sides is a promising direction of research but is under-explored here. We leave this for further investigation.

**Acknowledgements**    This work has been supported by Hong Kong Research Grant Council - Early Career Scheme (Grant No. 27209621), General Research Fund Scheme (Grant No. 17202422), RGC Theme-based research (T45-701/22-R) and RGC Matching Fund Scheme (RMGS). Part of the described research work is conducted in the JC STEM Lab of Robotics for Soft Materials funded by The Hong Kong Jockey Club Charities Trust.

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

## A A Heuristic Baseline for Co-occurrence Discovery

In this section, we introduce the baseline method used for ablation study in Table 4b in more detail. This baseline is adapted from a recently proposed image co-segmentation method ReCo [41]. As shown in Figure 6, it basically consists of four steps to identify the co-occurring object in the query image: First, it estimates pair-wise region similarity between region proposals of the query image and support images, which is the same as CoDet. This yields a similarity matrix $\mathbf{S} \in \mathbb{R}^{n \times m \times n}$, where $n$ stands for the number of proposals per image, and $m$ stands for the number of support images. Second, it applies a max operator on the last dimension of $\mathbf{S}$, which serves to find the nearest neighbor in each support image for each region proposal in the query image. This reduces $\mathbf{S}$ to an $n \times m$ matrix. Third, it applies a mean operator on the second dimension of $\mathbf{S}$ to derive the average support that each proposal has among the support images. Finally, it identifies the co-occurring object as the one with the highest average maximum similarity (support) among support images, by applying an argmax operator on the first dimension of $\mathbf{S}$.

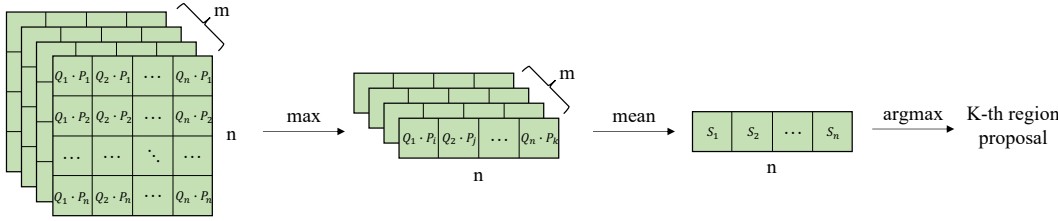

Figure 6: **Illustration of the baseline method for co-occurrence discovery**. $Q$ and $P$ are region proposals in the query image and support images, respectively. $S$ is the averaged maximum similarity score across support images.

## B Further Analysis on Different Alignment Strategies

Complementing the discourse in Section 4.5, we further delineate the performance of different alignment strategies with respect to novel category $AP_{50}$ on OV-COCO in Figure 7. It can be seen that strategies based on region-region similarity or hand-crafted rules (max-size) show steady improvement in novel object recognition across training, whereas the performance of region-word similarity-based method is highly unstable and even decreases in the early stage. A possible explanation is that solely relying on region-word similarity to align regions and words may be more susceptible to errors in pseudo-labels. For instance, if the model incorrectly matches the text label 'seagull' with the object 'dove' at the initial phase, its supervision signal would pull the two closer in the shared feature space. This negative feedback could directly harm the following pseudo-labeling process, thus, there is a higher probability for the model to make the same mistake.

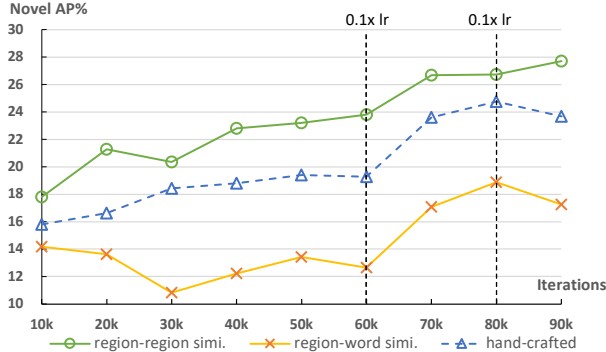

Figure 7: Performance of different alignment strategies at discrete training stages on OV-COCO.

# C Visualization on OV-LVIS and OV-COCO

We visualize more detection results of CoDet in Figure 8 and Figure 9. On OV-LVIS, we can see that CoDet successfully detects many rare objects, *e.g.*, gas mask, puffin, horse buggy, heron, satchel, and so on (Figure 8). This validates that CoDet can efficiently leverage web-crawled image-text pairs to learn open-word knowledge for novel object recognition. On OV-COCO, our method continues to demonstrate strong open-vocabulary capability and correctly detects some hard samples, *e.g.*, the occluded 'tie' and 'elephant' (upper left of Figure 9). Nevertheless, we also notice that the prediction scores for novel categories are generally lower than base categories, which suggests the model is biased towards base classes in OV-COCO. Such tendency to overfit base categories is also observed in other works [59, 26, 50], due to the small training vocabulary of OV-COCO. We believe adopting tricks like focal loss could alleviate this issue and further benefit our method.

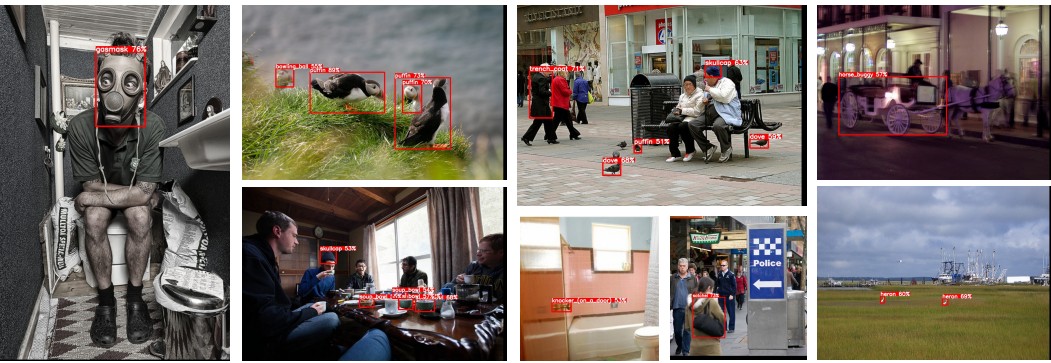

Figure 8: **Visualization of prediction results by CoDet on OV-LVIS**. For clarity, we only show results for novel categories.

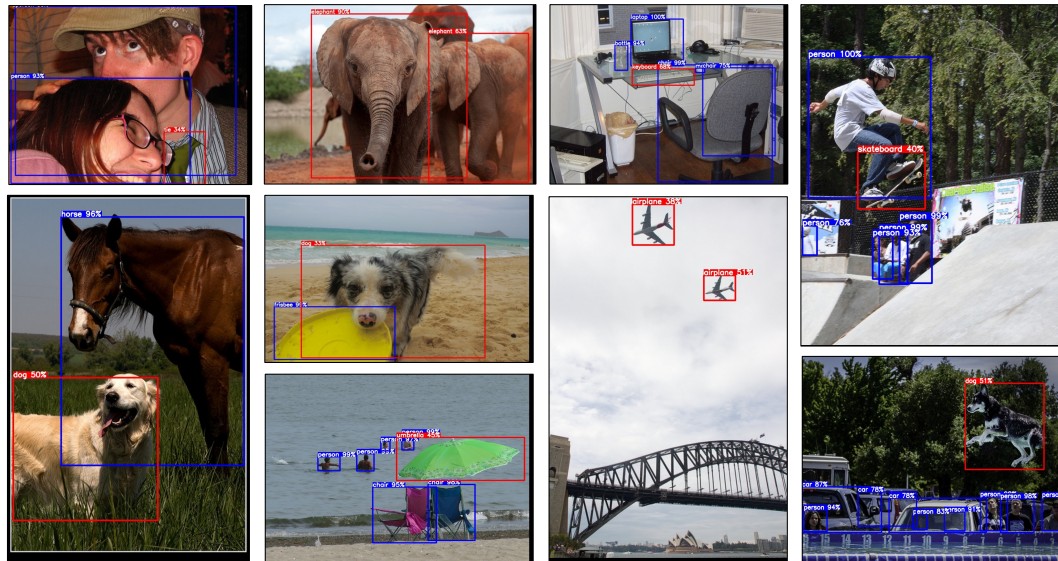

Figure 9: **Visualization of prediction results by CoDet on OV-COCO**. Red boxes are for novel categories, while blue boxes are for base categories.

# D Implementation Details

Table 6 lists the detailed hyper-parameter configuration used for our OV-LVIS and OV-COCO experiments. We follow Detic [61] to use low input resolution and large batch size for caption data to achieve better trade-off between efficiency and performance.

Table 6: **Hyper-parameter configuration of CoDet.** LSJ stands for large scale jittering [16]. Resolution refers to the resized short side length of input images.

| Configuration | OV-LVIS | OV-COCO |
|---|---|---|
| Optimizer | AdamW | SGD |
| Gradient clipping | True | True |
| Learning rate (LR) | 2e-4 | 2e-2 |
| Total iterations | 90k | 90k |
| Warmup iterations | 1k | – |
| Step decay factor | – | $0.1\times$ |
| Step decay schedule | – | [60k, 80k] |
| Data augmentation | LSJ | none |
| Batch size (detection) | 8 | 2 |
| Batch size (caption) | 32 | 8 |
| Detection/Caption data ratio | 1:4 | 1:4 |
| Federated loss [62] | True | False |
| Repeat factor sampling | True | False |
| $\mathcal{L}_{\text{region-word}}$ weight | 0.2 | 0.1 |
| $\mathcal{L}_{\text{image-text}}$ weight | 0.2 | 0.1 |

