# OpenReview forum: "CoDet: Co-occurrence Guided Region-Word Alignment for Open-Vocabulary Object Detection"
_NeurIPS.cc/2023/Conference — NeurIPS 2023 poster_

### Official Review · Reviewer_e66K · 2023-07-04

**Soundness:** 3 good
**Presentation:** 4 excellent
**Contribution:** 3 good
**Rating:** 6
**Confidence:** 3

**Summary:**

The paper presents CoDet, a novel approach for region-word alignment in vision-language representations for open-vocabulary object detection. Unlike existing methods that rely on pre-trained or self-trained models, CoDet reformulates alignment as a co-occurring object discovery problem. By grouping images that mention the same concept, CoDet establishes correspondences between shared concepts and common objects through co-occurrence, enabling it to leverage region-region correspondences across images for object discovery and open-vocabulary supervision. Experimental results demonstrate that CoDet consistently outperforms state-of-the-art methods in detecting novel objects, while also showcasing scalability with visual representations, indicating its potential for benefiting from advancements in visual foundation models.

**Strengths:**

+  In terms of originality, the authors propose a novel approach, CoDet, which reformulates region-word alignment as a co-occurring object discovery problem, diverging from the reliance on pre-trained or self-trained vision-language models. This unique perspective brings a fresh perspective to the field.
+ The quality of the paper is commendable, as the authors provide a clear description of the proposed approach, detailing how CoDet groups images based on shared concepts, leverages co-occurrence for region-region correspondences, and utilizes open-vocabulary supervision. The experimental results consistently outperform state-of-the-art methods, validating the effectiveness of CoDet in detecting novel objects.
+ The clarity of the paper is also notable, as the authors provide a concise and coherent presentation of their work, making it easily understandable to readers.
+ The significance of the paper lies in its potential impact on the vision-language representation field. By addressing the limitations of existing methods, CoDet opens up new possibilities for reliable region-word alignment and object-level vision-language representations, which can benefit various applications such as open-vocabulary object detection.

**Weaknesses:**

- The paper would benefit from a more explicit discussion of its differences and a comparative analysis with related work [a]. By highlighting the distinctions between the proposed CoDet approach and the existing method [a], the authors can provide a clearer understanding of the unique contributions and advantages of their approach.
[a] Aligning Bag of Regions for Open-Vocabulary Object Detection. CVPR, 2023.
- The absence of VLDet in Tables 2 and 3 raises concerns and it is important for the authors to provide an explanation for its omission. Including VLDet in the comparative analysis is crucial to assess its performance against the proposed CoDet approach and other existing methods.

**Questions:**

Please refer to the weaknesses mentioned above.

**Limitations:**

The authors have adequately addressed the limitations

---

> ### Author Rebuttal · Authors · 2023-08-10
>
> Dear Reviewer e66K,
> Thank you for your appreciation of our approach and constructive comments. We address your comments below.
> 1. Differences between Aligning Bag of Regions for Open-Vocabulary Object Detection.
>   - Thanks for pointing out this missing related work. We will add the discussion of BARON[1] into the Related Work section of our paper.
>   - BARON and CoDet might look similar as both works take advantage of "co-occurrence". However, the meaning of "co-occurrence" in CoDet is quite different from that in BARON. In CoDet, "co-occurrence" refers to the existence of the same object class across different images. While in BARON, "co-occurrence" refers to the existence of different object classes within the same image. Such a difference is further reflected in the motivation. BARON proposes to align "bag of regions" with "bag of concepts", while individual region-word alignment is not a main focus. In contrast, CoDet is making orthogonal efforts to discover single region-word pairs.
>   - Moreover, BARON is a distillation-based method, which still relies on a teacher VLM and inherits constraints from the image-level pre-trained VLM. While CoDet overcomes such dependency by introducing a new region-word alignment mechanism.
>   - CoDet performs on par with BARON on OV-LVIS benchmark (22.7 v.s. 22.7 mAPr), and outperforms BARON in transfer detection from OV-LVIS to COCO and Objects365. The transfer detection results are presented below. Notably, the results of CoDet on Objects365 are different from those reported in the paper because here we use Objects365 v2 val for evaluation instead of Objects365 v1, to keep in accordance with BARON. We directly cite the number reported in BARON.
> | Dataset     |          |   COCO   |          |          | Obj365 |          |
> |:------:|:--------:|:--------:|:--------:|:--------:|:--------:|:--------:|
> |  Method      |    AP    |   AP$_{50}$   |   AP$_{75}$   |    AP     |   AP$_{50}$   |   AP$_{75}$   |
> | BARON | 36.2 | 55.7 | 39.1 | 13.6 | **21.0** | 14.5 |
> | CoDet | **38.5** | **55.8** | **41.5** | **14.5** | 20.6 | **15.7** |
>
> 2. The absence of VLDet in Tables 2 and 3.
>   - Thanks for your suggestions. We will add VLDet to Table 2 & 3 for comparison.
>   - We primarily benchmarked and analyzed our method on OV-LVIS in the paper as we believe OV-LVIS results are more representative than OV-COCO results because:
>     - OV-LVIS has many more novel categories for evaluation (337 in OV-LVIS v.s. 17 in OV-COCO)
>     - OV-LVIS setting uses large-scale web-crawled caption data for training, which better simulates real-word practice (OV-COCO setting only uses 120K human-annotated caption data for training).
>   - It is noteworthy to point out that VLDet has superior performance over CoDet on OV-COCO (32.0 v.s. 30.6 AP50 on novel classes). We have already presented an analysis of potential causes in our paper. This could be attributed to:
>     - The human-curated bias in COCO Caption data distribution. As analysed in lines 317-323, concepts in COCO Caption images are highly concentrated, which incurs many hard negatives for identifying co-occurring objects. But we believe this would not harm the generailty of our method as we show CoDet works well on web-crawled data, which is a more practical setting and can easily be scaled up.
>     - VLDet has more aggressive trade-offs between novel AP and base AP on OV-COCO. CoDet actually has a higher base AP (52.3 v.s. 50.6) and overall AP (46.6 v.s. 45.8) than VLDet. This trade-off typically happens in methods trained on the detection and caption data simultaneously, e.g., Detic, VLDet, CoDet.
>   - We did not include VLDet for comparison in Table 3 because the original paper did not report their transfer detection results on OV-LVIS to Objects365 and COCO. Here for comparison, we use the officially released checkpoint of VLDet for evaluation.
>   - It can be seen that CoDet outperforms VLDet on large-sclae Objects365 v2 transfer detection, showing its higher capability for detecting a wide range of novel concepts. However, an interesting finding is that VLDet has stronger transfer detection results on COCO, compared with CoDet and BARON. This is not as expected, given that VLDet has a bit lower mAPr on OV-LVIS than CoDet and BARON. It seems VLDet has unique advantages on common object detection. The deep reason behind it is still unclear and worth further studies.
> | Dataset          |          |   COCO   |          |          | Obj365 |          |
> |:------:|:--------:|:--------:|:--------:|:--------:|:--------:|:--------:|
> |  Method      |    AP    |   AP$_{50}$   |   AP$_{75}$   |    AP     |   AP$_{50}$   |   AP$_{75}$   |
> |  VLDet | **39.7** | **56.9** | **43.2** |   12.8   |    18.0    |   13.9   |
> |  CoDet |   38.5   |   55.8   |   41.5   | **14.5** |  **20.6**  | **15.7** |
>
> &nbsp;
> &nbsp;
>
> References:
>
> [1] Aligning Bag of Regions for Open-Vocabulary Object Detection

---

### Official Review · Reviewer_Z3qw · 2023-07-07

**Soundness:** 3 good
**Presentation:** 3 good
**Contribution:** 2 fair
**Rating:** 5
**Confidence:** 3

**Summary:**

This paper proposes CoDet, a novel approach for open-vocabulary object detection that reformulates region-word alignment as a co-occurring object discovery problem. The approach groups images that mention the same concept in their captions, leveraging region-region correspondences to discover the common object and adopt the shared concept as category label for open-vocabulary supervision. Experimental results demonstrate that CoDet consistently outperforms state-of-the-art methods in detecting novel objects.

**Strengths:**

1. Introducing a new method, CoDet, for solving the open-vocabulary object detection problem, which reformulates region-word alignment as a co-occurring object discovery problem, achieving more accurate localization and better generalization ability.
2. Experimental results demonstrate that CoDet outperforms existing state-of-the-art methods in detecting novel objects and exhibits strong scalability, benefiting from advancements in visual foundation models.
3. The authors provided open-source code, which ensures the reproducibility of experiments and promotes further research.
4. The paper is well-organized, with a clear structure and logical flow.


**Weaknesses:**

1. The method is simple, and prototype-based solutions are common in other tasks. I am slightly concerned about the novelty of the approach.


2.  There are  fairness concerns in this paper. See Questions.


**Questions:**

1. The fairness for comparison cannot be promised. For LVIS, CenterNet2 is used in this paper. But in previous methods, FasterRCNN is usually used.

2. Some implement details are confusing. Why using CenterNet2 [62] with ResNet50 as the backbone for LVIS?. But for OV-COCO, Faster R-CNN with a ResNet50-C4 backbone is adopted.  Is this fair to compare with previous methods? The same backbone should be used for fair comparison.

3. Some OVD methods are not compared in this paper. e.g., "Open-vocabulary detr with conditional matching."




**Limitations:**

The author have addressed the limitations.

---

> ### Author Rebuttal · Authors · 2023-08-10
>
> Dear Reviewer Z3qw,
>
> We really appreciate your comments. We hope our response can address your concerns and clarify our contribution.
> 1. Novelty concern
>   - We believe method simplicity should not harm paper novelty. In contrast, without redundant design, simple and effective methods usually provide clear and concise insights into the research problem.
>   - Besides, CoDet contributes a new perspective to address the OVD problem. As confirmed by all other reviewers, the idea of reformulating region-word alignment as a co-occurring object discovery problem is unique and interesting, which overcomes the reliance on pre-trained or self-trained vision-language models for alignment. Reviewer 2fPU further confirms this idea has reference values for future works within and beyond OVD research.
>   - While prototypes are commonly used in other tasks like few-shot learning, CoDet is fundamentally different from these methods in the way and insight of constructing and using prototypes. For prototype construction, rather than relying on annotated samples, we introduce similarity-based prototype synthesis to automatically discover prototypes for target objects. Besides, CoDet does not directly use prototypes for classification. Instead, prototypes are only intermediate products to learn vision-language alignment. For inference, CoDet no more needs prototypes.
> 2. Fairness concern in comparison
>   - First, we would like to point out that "using CenterNet2 for LVIS experiments and Faster R-CNN for COCO experiments" is not an "innovation" of this paper. As put in line 234, we follow exactly the same setting as Detic [1], a well-known OVD work (167 citations by now). VLDet [2] also adopted this setting.
>   - Second, we choose to follow the Detic setting because it is a mature setting that has been widely acknowledged by the community. It can be seen that most of, if not all, recent OVD works include Detic results in their comparisons, e.g., PromptDet [3], F-VLM [4], VLDet, CORA [5].
>   - Third, it is hard to find a universal standard for implementation. Discrepancy in implementation is common among different OVD works. For instance, ViLD [6] uses a 32x training schedule (CoDet only uses 4x), DetPro [7] and BARON [8] use detection-specialized pretraining SoCo[9] for weight initialization, OV-DETR [10] uses Deformable-DETR, not mention that many works do not even follow the strict open-vocabulary setting.
> 3. "Open-vocabulary detr with conditional matching" is not compared in this paper.
>   - OV-DETR has already been included in comparison (Please see Table 1 & 2).
>
> &nbsp;
> &nbsp;
> &nbsp;
>
> References:
>
> [1] Detecting Twenty-thousand Classes using Image-level Supervision
>
> [2] Learning Object-Language Alignments for Open-Vocabulary Object Detection
>
> [3] PromptDet: Towards Open-vocabulary Detection using Uncurated Images
>
> [4] F-VLM: Open-Vocabulary Object Detection upon Frozen Vision and Language Models
>
> [5] CORA: Adapting CLIP for Open-Vocabulary Detection with Region Prompting and Anchor Pre-Matching
>
> [6] Open-vocabulary Object Detection via Vision and Language Knowledge Distillation
>
> [7] Learning to Prompt for Open-Vocabulary Object Detection with Vision-Language Model
>
> [8] Aligning Bag of Regions for Open-Vocabulary Object Detection
>
> [9] Aligning pretraining for detection via object-level contrastive learning.
>
> [10] Open-vocabulary detr with conditional matching.

---

> > ### Comment · Reviewer_Z3qw · 2023-08-17
> >
> > The authors has addressed my concerns. Although I still think the method of this paper is simple, it is a good paper that provides a new perspective. After reading the rebuttal and comments from other reviewers, I upgraded the rating.

---

> > > ### Author Response · Authors · 2023-08-17
> > > **Thanks for Your Support of Our Work**
> > >
> > > Thank you very much for taking the time to reconsider our paper submission. We appreciate you engaging with us in the rebuttal process, thoughtfully considering our responses, and agreeing to upgrade the rating of our paper. We are grateful for your open-mindedness and willingness to re-evaluate our work.

---

### Official Review · Reviewer_CkPP · 2023-07-11

**Soundness:** 3 good
**Presentation:** 3 good
**Contribution:** 3 good
**Rating:** 6
**Confidence:** 3

**Summary:**

The paper proposes a novel perspective in discovering region-word correspondence from image-text pairs, which bypasses the dependence on a pre-aligned vision-language space by reformulating region-word alignment as a co-occurring object discovery problem. An open-vocabulary object detection framework named CoDet is built and achieves state-of-the-art performance across multiple standard benchmarks. The proposed method also demonstrates the effectiveness of visual guidance in region-word alignment. Experimental results validate the effectiveness of the method.

**Strengths:**

The paper is well-written and easy to understand. The idea of using co-occurrent regions as visual guidance is interesting. Experimental results are promising.

**Weaknesses:**

1. During prototype construction, how to ensure that the prototype does not contain noisy information such as hard negative samples.

2. In line 16, the authors claim that the proposed method can benefit from advancements in visual foundation models. However, no further experiments about different visual foundation models are carried out.

**Questions:**

Please see above.

**Limitations:**

The authors addressed the limitations.

---

> ### Author Rebuttal · Authors · 2023-08-10
>
> Dear Reviewer CkPP,
>
> Thanks so much for your constructive comments and support for acceptance. We hope our response can address your concerns.
> 1. How to ensure that the prototype does not contain noisy information such as hard negative samples?
>   - This is a very good question. Since we are aggregating all region proposals of an image into a prototype, we are not able to ensure 100% purity of the prototype. But we found our learning-based prototype synthesis is relatively robust to the noisy samples. From our observation, most of the proposals got weights smaller than 0.01 in prototype synthesis, which suggests the algorithm automatically learns from the similarity matrix to suppress noisy information.
>   - As for hard negative samples, text guidance serves as an effective method to filter them (only proposals corresponding to the target concept will get a high similarity score). Figure 5 gives an example of how text guidance filters hard negatives.
> 2. No further experiments on visual foundation models.
>   - We would like to clarify that the original content in line 16 is "CoDet exhibits its potential to benefit from advancements in visual foundation models", which is based on two facets:
>     - Theoretically, CoDet mostly relies on visual correspondences to identify co-occurring objects, thus robust visual features from foundation models would benefit CoDet in finding more accurate co-occurrences.
>     - Empirically, our experiments show that adopting stronger visual backbones (Resnet50 -> SwinB, Table 1) leads to significant performance boost (+6.7 mAP on novel classes).
>
>     For the aforementioned reasons, we believe using stronger visual foundation models as backbone would possibly continue to bring further performance gains.
>   - We totally agree that adding experiments on visual foundation models is a great idea to improve our work. Therefore, we are running OV-LVIS experiments using EVA-02 [1] pre-trained model, which achieves 90.0% top-1 accuracy on ImageNet-1k val. As this experiment takes a long time, we will post the results later during the author-reviewer discussion period.
>
> &nbsp;
> &nbsp;
>
> References:
>
> [1] EVA-02: A Visual Representation for Neon Genesis

---

> > ### Author Response · Authors · 2023-08-17
> > **Experiment Results on Visual Foundation Models**
> >
> > Here we report the results of CoDet using EVA02-L as the backbone on OV-LVIS, which further verifies that CoDet can consistently benefit from stronger visual representations. In comparison with F-VLM [1] which uses the pre-trained CLIP visual encoder as backbone, CoDet achieves superior performances with fewer parameters.
> >
> > | Method | Backbone    | Params. | mAPr     | mAPc     | mAPf     | mAP      |
> > |--------|-------------|------------|----------|----------|----------|----------|
> > |           | R50    | 25M        | 22.7     | 30.3     | 34.7     | 30.7     |
> > | CoDet | Swin-B      | 88M        | 29.4     | 39.5     | 43.0     | 39.2     |
> > |           | **EVA02-L** | **304M**   | **35.2** | **49.1** | **49.2** | **46.7** |
> > | F-VLM  | R50x64      | 420M       | 32.8     | --       | --       | 34.9     |
> >
> > &nbsp;
> > &nbsp;
> >
> > [1] F-VLM: Open-Vocabulary Object Detection upon Frozen Vision and Language Models

---

> > > ### Comment · Reviewer_CkPP · 2023-08-19
> > >
> > > Thanks for the response. The authors have addressed my concerns. Therefore, I will keep my rating.

---

### Official Review · Reviewer_2fPU · 2023-07-13

**Soundness:** 3 good
**Presentation:** 2 fair
**Contribution:** 3 good
**Rating:** 7
**Confidence:** 4

**Summary:**

This paper argues that existing OVD work relies heavily on visual language pre-training tasks, and current methods cannot provide more fine-grained cross-modal alignment information for OVD, resulting in limited performance on open vocabulary detection tasks.
To this end, the authors propose to adopt co-occurrence guided region-word alignment for the OVD task.
This method is novel and interesting, and has a good reference value for fields such as OVD and open world segmentation.
Overall, the writing of this article is clear and the content is comprehensive. But I think the author still needs a clearer motivation, formulation and explanation for how to discover the co-occurrence proposal in the image.

**Strengths:**

As expressed in the summary, the ideas in this paper are novel and interesting, and the writing is clear and complete.

**Weaknesses:**

In this paper, the motivation and solution of how to obtain the co-occurrence proposal corresponding to the text from the mini-group image is not clear. For example,
- How the author avoids the problem of large differences in the visual appearance of target objects in different images under the same concept mentioned in line 170.
- In addition, the author's description of the change in the shape of the similarity matrix in figure2 is too late (in line 185), and it may actually be more appropriate to arrange it in line 166. This caused some difficulty in understanding.

**Questions:**

1.How the author avoids the problem of large differences in the visual appearance of target objects in different images under the same concept mentioned in line 170.

2.In line 189, why should the co-occurrence region appear in the last dimension of S?

3.In line 177, the text feature should be an embedding of a sentence, why does the author say that the relative size of text features in different dimensions indicates their relative importance in concept classification?



**Limitations:**

The authors provide an objective discussion of the limitations of the article.

---

> ### Author Rebuttal · Authors · 2023-08-10
>
> Dear Reviewer 2fPU,
>
> Thank you very much for your constructive comments which helped improve our manuscript, as well as your support for acceptance. We hope our response can address your concerns.
> 1. Dealing with large intra-class variance of object appearances
>   - We adopt text guidance to suppress the impacts of intra-class visual variations.
>   - First, we observe that intra-class variations are typically reflected in particular visual attributes. For example, some object classes have large variances in color, and some have large variances in texture. Those visual attributes of large intra-class variance are naturally considered uninformative in the classification of that class.
>   - Second, in the shared vision-language space, the class text embeddings are forced to have high similarity scores with visual features from the same class, agnostic to the visual feature variance during contrastive learning. This encourages the text embeddings to assign high weights to dimensions corresponding to visual attributes of low intra-class variance, and assign low weights to those corresponding to attributes of high variance.
>   - We thus leverage the learned class text embeddings for feature selection (see Equation) to emphasize invariant features and suppress features with high variations. For instance, suppose the visual feature of "dog" has high variance at dimension 0 but low variance at dimensions 1 & 2, we would expect the text embedding of "dog" to be something like [0.1, -0.6, 0.7], where 0.1 indicates the visual attribute at dimension 0 is unimportant in the classification of "dog". Based on this observation, we use the absolute value of text embeddings to reweight similarity estimation (Equation 1). Thus, intra-class variations are suppressed by small weights from corresponding text embeddings.
> 2. Move description of the similarity matrix calculation upward
>   - Thanks for your valuable feedback to enhance the work! We agree that it would be more logically coherent to put the introduction of "similarity-based prototype discovery" before "text guidance" as the former constitutes the major part of our method. We will rearrange the layout of the paper as suggested.
> 3. In line 189, why should the co-occurrence region appear in the last dimension of S?
>   - This is because the last dimension of S essentially includes the similarity scores between the query proposal and region proposals from the support images. Among these scores, the co-occurring region proposals are characterized by having high values.
> 4. How does the relative magnitude of text features at different dimensions indicate their relative importance in concept classification?
>   - We hope our discussion in question 1 addresses this question as well.

---

> > ### Comment · Reviewer_2fPU · 2023-08-21
> > **Official Comment by Reviewer 2fPU**
> >
> > Thank you very much for your detailed explanation of my concerns.

---

### Official Review · Reviewer_Puz3 · 2023-07-26

**Soundness:** 3 good
**Presentation:** 2 fair
**Contribution:** 3 good
**Rating:** 5
**Confidence:** 4

**Summary:**

In this paper, they propose CoDet, a novel approach that overcomes the reliance on pre-aligned vision-language space by reformulating
region-word alignment as a co-occurring object discovery problem. Specially, CoDet groups images that mention the same concept in their captions, which brings a natural correspondence between the shared concept and the common objects within the group through co-occurrence. Experimental results demonstrate that CoDet consistently outperforms state-of-the-art methods in detecting novel objects.

**Strengths:**

1. This paper propose an interesting idea to reformulate region-word alignment as a co-occurring object discovery problem.
2. Experiments show its effectivenesses.

**Weaknesses:**

1. Experimentation Missing, Need to compare with the fine-grained word-region alignment method proprosed in DetCLIPv2[1], another OVD method try to learn with image-text pair more efficiently with region-word loss.
2. Some more questions, see the questions.

[1] DetCLIPv2: Scalable Open-Vocabulary Object Detection Pre-training via Word-Region Alignment

**Questions:**

1. The image-text pairs currently used are relatively clean. It is hypothetical that if data with larger noise, such as YFCC, are used, the results might differ. In their images, the content described in many texts does not align with the images. Would the algorithm proposed in the paper have limitations in such cases? Furthermore, how efficient is the proposed algorithm? For instance, when expanded to larger datasets like YFCC26M, how does it affect the increase in training time and the stability of performance improvement?
2. Suppose there are two noun  that your models have never learned similar concepts (where text guidance is ineffective), and the objects corresponding to these two nouns always appear simultaneously, such as eyes and eyelashes. Would it be difficult for this framework to distinguish between them, or is all existing methods fundamentally incapable of solving such issues?
3. Regarding the technical details in lines 160-161, it appears that multiple prototypes and text losses are computed simultaneously in a single forward pass, followed by iterative selection of the query image, correct?
4. How sensitive is this framework to RPN? For example, in the case of Figure 2, if the RPN proposes a dog's head, is there a possibility that the dog's head might be recognized as a complete dog?

**Limitations:**

The limitations are addressed.

---

> ### Author Rebuttal · Authors · 2023-08-10
>
> Dear Reviewer Puz3,
>
> Thanks a lot for your insightful reviews and support for our work! We hope our response can address your questions.
> 1. Need comparison with DetCLIPv2
>   - Thanks for pointing out this missing **contemporary work**. We will add the discussion of DetCLIPv2 into the Related Work section of our paper.
>   - CoDet and DetCLIPv2 are not readily comparable at this point due to different training and evaluation protocols (DetCLIPv2 follows the setting in GLIP [1], while CoDet follows the setting in ViLD [2]). In detail, for training, DetCLIPv2 uses 5x more data (roughly 15M image-text pairs and 1.44M region-text pairs) than CoDet. For evaluation, DetCLIPv2 uses the full LVIS rather than OV-LVIS which splits LVIS into seen/unseen categories.
>   - We follow the setting in ViLD due to computational resource constraints. Moreover, as there is no open-sourced code of DetCLIPv2, reproducing and evaluating DetCLIPv2 using the same data as CoDet is not feasible, given limited rebuttal time.
>   - Finally, from the high-level insight, our effort in reformulating region-word alignment as co-occurring object discovery are orthogonal to the efforts in DetCLIPv2, which explores aligning regions and words directly.
> 2. Dealing with noisier image-text pairs (e.g., YFCC)
>   - This is quite an intriguing point of observation. Typically, poor quality of training image-text pairs degrades the model performance. This is **a common issue** for pseudo-label-based OVD methods, e.g., VLDet [3] also has related discussions in Appendix 4. Failure Cases.
>   - CoDet is designed to be relatively robust to such noises. The minimum requirement for discovering co-occurring objects is that the target object exists in the majority of images within a concept group, though not necessarily in all images.
>   - In our early experiments, we attempted to filter out noisy samples, e.g., if an image did not have proposals with sufficiently close neighbors across the majority of images in the mini-group, it will be considered not including the target concept, thus being removed. This design didn't bring notable performance gains yet introduced unnecessary complexities. Thus, we discarded this design. But in case of noisier image-text pairs are used for training, e.g., YFCC, this in-place filtering method might prove beneficial. We may explore this in the future.
> 3. Training efficiency.
>   - The training time of CoDet grows linearly with the amount of training data. This means CoDet can be easily scaled to web-scale data.
> 4. Stability of performance improvement when scaled to larger datasets.
>   - This is a good point worth future studies. In our experiments with the OV-COCO setting (small-scale) and the OV-LVIS setting (large-scale), we see a significant performance improvement over baselines in both settings. This is a positive sign that CoDet can scale up to larger datasets.
> 5. How to distinguish co-existing concepts?
>   - Yes, this can be a systematic failure for existing OVD methods to distinguish co-existing concepts that have never been seen before. Co-existing concepts will incur high ambiguities in associating concepts and regions. For example, in methods based on region-word similarity matching such as VLDet, the ambiguity lies in associating regions with corresponding text supervision. For our method, the ambiguity lies in associating objects discovered in a group with corresponding co-occurring concepts.
>   - But we think this is not a failure of algorithms but the failure of data -- even humans could hardly distinguish two unseen concepts that "always appear concurrently" without any prior knowledge of them. We believe a solution to this problem is to scale up training data. Given sufficient data, hopefully, we can find samples that break up the concurrency of the two concepts. Another is to inject priors for learning more discriminative embeddings of the two concepts.
> 6. Clarification on technical details in lines 160-161
>   - The query image is set before computing prototypes and text losses. The full training pipeline can be summarized as follows:
>   - sample images from the same concept group -> extract region proposals for each image -> iteratively set an image as the query image -> compute similarity matrix between proposals of the query image and support images -> synthesize a prototype for the query image -> compute text (classification) loss
>   - Note that although we use "iteratively" to describe this process of query image selection, prototype synthesis is actually independent of each other. This means in practice, prototype synthesis can be executed simultaneously for efficiency.
> 7. Is there a possibility that the dog's head might be recognized as a complete dog?
>   - There are two possible scenarios. If the image only contains a part of the dog, e.g., dog head, CoDet will recognize the dog head as a dog, which is expected behavior. If the image contains a complete dog, the proposal of dog head will be suppressed by the proposal of a complete dog in NMS, which is often the case. Therefore, it is okay for the RPN to generate some part-level proposals, as long as we can handle them in post-processing, i.e. NMS. Moreover, when conducting visualization, we did not find the misclassification of object parts as objects to be a systematic issue of CoDet.
>
> &nbsp;
> &nbsp;
>
> References:
>
> [1] Grounded Language-Image Pre-training
>
> [2] Open-vocabulary Object Detection via Vision and Language Knowledge Distillation
>
> [3]  Learning Object-Language Alignments for Open-Vocabulary Object Detection

---

> > ### Comment · Reviewer_Puz3 · 2023-08-21
> > **Response to the author**
> >
> > Thank you very much for your detailed explanation of my concerns. I keep my initial score.

---

### Author Rebuttal · Authors · 2023-08-10

Dear Reviewers and ACs:

Thank you so much for your time and efforts in assessing our paper. Hope our rebuttal has addressed your concerns. We are happy to further discuss with you if there are still other concerns. Thanks for helping improve our paper.

Best regards, Paper 2038 Authors

---

### Author Response · Authors · 2023-08-21
**Welcoming further discussions**

Dear Reviewers,

We sincerely thank you for your efforts in reviewing our paper and your suggestions for polishing the manuscript. As we are approaching the end of the discussion period, we would like to ask whether there are any remaining concerns regarding our paper or our response. We are happy to answer any further questions.

If you find our responses have addressed all of your concerns, we would be much grateful if you could kindly consider raising the rating of our paper during your final evaluation.

---

### Decision · Program_Chairs · 2023-09-21

**Decision:**

Accept (poster)

**Comment:**

This paper proposes a co-occurrence guided region-word alignment approach for Open-Vocabulary Object Detection tasks. All reviewers give positive scores after the rebuttal. The newly added results on LVIS and obj365 look convincing. The AC decides to accept it.